# Uncertainty in Neural Processes

## Abstract

We explore the effects of architecture and training objective choice on amortized posterior predictive inference in probabilistic conditional generative models. We aim this work to be a counterpoint to a recent trend in the literature that stresses achieving good samples when the amount of conditioning data is large. We instead focus our attention on the case where the amount of conditioning data is small. We highlight specific architecture and objective choices that we find lead to qualitative and quantitative improvement to posterior inference in this low data regime. Specifically we explore the effects of choices of pooling operator and variational family on posterior quality in neural processes. Superior posterior predictive samples drawn from our novel neural process architectures are demonstrated via image completion/in-painting experiments.

## 1 Introduction

What makes a probabilistic conditional generative model *good*? The belief that a generative model is good if it produces samples that are indistinguishable from those that it was trained on (Hinton, 2007) is widely accepted, and understandably so. This belief also applies when the generator is conditional, though the standard becomes higher: conditional samples must be indistinguishable from training samples for each value of the condition.

Consider an amortized image in-painting task in which the objective is to fill in missing pixel values given a subset of observed pixel values. If the number and location of observed pixels is fixed, then a good conditional generative model should produce sharp-looking sample images, all of which should be compatible with the observed pixel values. If the number and location of observed pixels is allowed to vary, the same should remain true for each set of observed pixels. Recent work on this problem has focused on reconstructing an entire image from as small a conditioning set as possible. As shown in Fig. 1, state-of-the-art methods (Kim et al., 2018) achieve high-quality reconstruction from as few as 30 conditioning pixels in a 1024-pixel image.

Our work starts by questioning whether reconstructing an image from a small subset of pixels is always the right objective. To illustrate, consider the image completion task on handwritten digits. A small set of pixels might, depending on their locations, rule out the possibility that the full image is, say, 1, 5, or 6. Human-like performance in this case would generate sharp-looking sample images for *all* digits that are consistent with the observed pixels (i.e., 0, 2-4, and 7-9). Observing additional pixels will rule out successively more digits until the only remaining uncertainty pertains to stylistic details. The bottom-right panel of Fig. 1 demonstrates this type of "calibrated" uncertainty.

We argue that in addition to high-quality reconstruction based on large conditioning sets, amortized conditional inference methods should aim for meaningful, calibrated uncertainty, particularly for small conditioning sets. For different problems, this may mean different things (see discussion in Section 3). In this work, we focus on the image in-painting problem, and define well calibrated uncertainty to be a combination of two qualities: high sample diversity for small conditioning sets; and sharp-looking, realistic images for any size of conditioning set. As the size of the conditioning set grows, we expect the sample diversity to decrease and the quality of the images to increase. We note that this emphasis is different from the current trend in the literature, which has focused primarily on making sharp and accurate image completions when the size of the conditioning context is large (Kim et al., 2018).

To better understand and make progress toward our aim, we employ posterior predictive inference in a conditional generative latent-variable model, with a particular focus on neural processes (NPs)

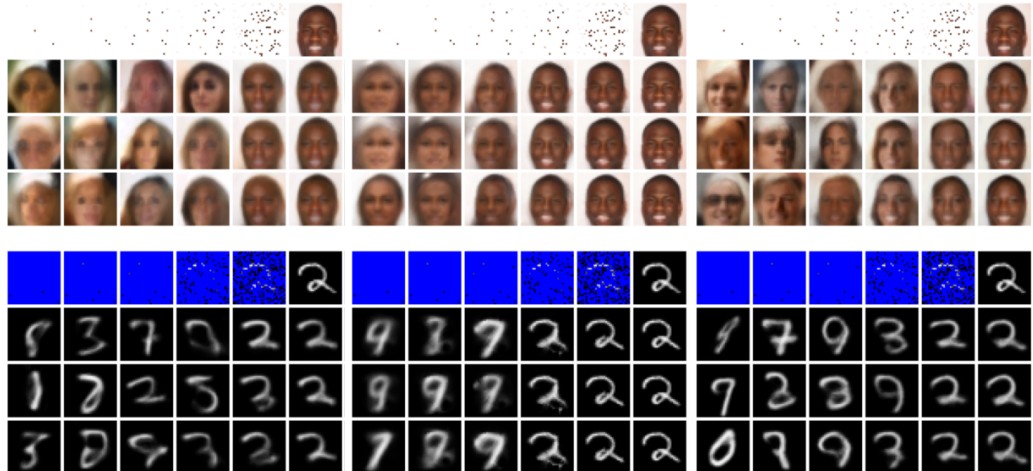

Figure 1: Representative image in-painting results for CelebA and MNIST. From left to right, neural process (NP) (Garnelo et al., 2018b), attentive neural process (ANP) (Kim et al., 2018), and ours. Top rows show context sets of given pixels, ranging from very few pixels to all pixels. In each panel the ground truth image (all pixels) is in the upper right corner. The rows correspond to i.i.d. samples from the corresponding image completion model given only the pixels shown in the top row of the same column. Our neural process with semi-implicit variational inference and max pooling produces results with the following characteristics: 1) the images generated with a small amount of contextual information are "sharper" and more face- and digit-like than NP results and 2) there is greater sample diversity across the i.i.d. samples than those from the ANP. This kind of "calibrated uncertainty" is what we target throughout.

(Garnelo et al., 2018a;b). We find that particular architecture choices can result in markedly different performance. In order to understand this, we investigate posterior uncertainty in NP models (Section 4), and we use our findings to establish new best practices for NP amortized inference artifacts with well-calibrated uncertainty. In particular, we demonstrate improvements arising from a combination of max pooling, a mixture variational distribution, and a "normal" amortized variational inference objective.

The rest of this paper is organized as follows. Section 2 and Section 3 present background material on amortized inference for generative models and calibrated uncertainty, respectively. Section 4 discusses and presents empirical evidence for how NP models handle uncertainty. Section 5 introduces our proposed network architecture and objective. Section 6 reports our results on the MNIST, FashionMNIST and CelebA datasets. Finally, Section 7 presents our conclusions.

## 2 AMORTIZED INFERENCE FOR CONDITIONAL GENERATIVE MODELS

Our work builds on amortized inference (Gershman & Goodman, 2014; Kingma & Welling, 2014), probabilistic meta-learning (Gordon et al., 2019), and conditional generative models in the form of neural processes (Garnelo et al., 2018b; Kim et al., 2018). This section provides background.

Let $(\boldsymbol{x}_{\mathcal{C}}, \boldsymbol{y}_{\mathcal{C}}) = \{(x_i, y_i)\}_{i=1}^{n}$ and $(\boldsymbol{x}_{\mathcal{T}}, \boldsymbol{y}_{\mathcal{T}}) = \{(x'_j, y'_j)\}_{j=1}^{m}$ be a context set and target set respectively. In image in-painting, the context set input $\boldsymbol{x}_{\mathcal{C}}$ is a subset of an image's pixel coordinates, the context set output $\boldsymbol{y}_{\mathcal{C}}$ are the corresponding pixel values (greyscale intensity or colors), the target set input $\boldsymbol{x}_{\mathcal{T}}$ is a set of pixel coordinates requiring in-painting, and the target set output $\boldsymbol{y}_{\mathcal{T}}$ is the corresponding set of target pixel values. The corresponding graphical model is shown in Fig. 2.

The goal of amortized conditional inference is to rapidly approximate, at "test time," the posterior predictive distribution

$$p_\theta(\boldsymbol{y}_{\mathcal{T}}|\boldsymbol{x}_{\mathcal{T}}, \boldsymbol{x}_{\mathcal{C}}, \boldsymbol{y}_{\mathcal{C}}) = \int p_\theta(\boldsymbol{y}_{\mathcal{T}}|\boldsymbol{x}_{\mathcal{T}}, z)p_\theta(z|\boldsymbol{x}_{\mathcal{C}}, \boldsymbol{y}_{\mathcal{C}})dz \ . \tag{1}$$

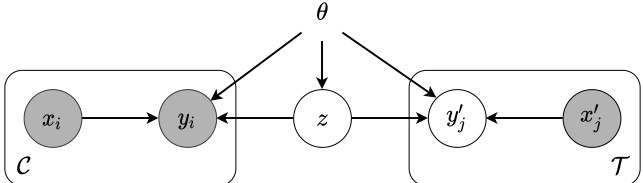

Figure 2: Generative graphical model for a *single* neural process task. $\mathcal{C}$ is the task "context" set of input/output pairs $(x_i, y_i)$ and $\mathcal{T}$ is a target set in which only the input values are known.

We can think of the latent variable $z$ as representing a problem-specific task-encoding. The likelihood term $p_\theta(\boldsymbol{y}_\mathcal{T}|\boldsymbol{x}_\mathcal{T}, z)$ shows that the encoding parameterizes a regression model linking the target inputs to the target outputs. In the NP perspective, $z$ is a function and Eq. (1) can be seen as integrating over the regression function itself, as in Gaussian process regression (Rasmussen, 2003).

**Variational inference**    There are two fundamental aims for amortized inference for conditional generative models: learning the model, parameterized by $\theta$, that produces good samples, and producing an amortization artifact, parameterized by $\phi$, that can be used to approximately solve Eq. (1) quickly at test time. Variational inference techniques couple the two learning problems. Let $\boldsymbol{y}$ and $\boldsymbol{x}$ be task-specific output and input sets, respectively, and assume that at training time we know the values of $\boldsymbol{y}$. We can construct the usual single-training-task evidence lower bound (ELBO) as

$$\log p_\theta(\boldsymbol{y}|\boldsymbol{x}) \geq \mathbb{E}_{\boldsymbol{z} \sim q_\phi(z|\boldsymbol{x},\boldsymbol{y})} \left[ \log \frac{p_\theta(\boldsymbol{y}|z,\boldsymbol{x})p_\theta(z)}{q_\phi(z|\boldsymbol{x},\boldsymbol{y})} \right] . \tag{2}$$

Summing over all training examples and optimizing Eq. (2) with respect to $\phi$ learns an amortized inference artifact that takes a context set and returns a task embedding; optimizing with respect to $\theta$ learns a problem-specific generative model. Optimizing both simultaneously results in an amortized inference artifact bespoke to the overall problem domain.

At test time the learned model and inference artifacts can be combined to perform amortized posterior predictive inference, approximating Eq. (1) with

$$p_\theta(\boldsymbol{y}_\mathcal{T}|\boldsymbol{x}_\mathcal{T}, \boldsymbol{x}_\mathcal{C}, \boldsymbol{y}_\mathcal{C}) \approx \int p_\theta(\boldsymbol{y}_\mathcal{T}|\boldsymbol{x}_\mathcal{T}, z)q_\phi(z|\boldsymbol{x}_\mathcal{C}, \boldsymbol{y}_\mathcal{C})dz . \tag{3}$$

Crucially, given an input $(\boldsymbol{x}_\mathcal{C}, \boldsymbol{y}_\mathcal{C})$, sampling from this distribution is as simple as sampling a task embedding $z$ from $q_\phi(z|\boldsymbol{x}_\mathcal{C}, \boldsymbol{y}_\mathcal{C})$ and then passing the sampled $z$ to the generative model $p_\theta(\boldsymbol{y}_\mathcal{T}|\boldsymbol{x}_\mathcal{T}, z)$ to produce samples from the conditional generative model.

**Meta-learning**    The task-specific problem becomes a meta-learning problem when learning a regression model $\theta$ that performs well on *multiple* tasks with the same graphical structure, trained on data for which the target outputs $\{y'_j\}$ are observed as well. In training our in-painting models, following conventions in the literature (Garnelo et al., 2018a;b), tasks are simply random-size subsets of random pixel locations $\boldsymbol{x}$ and values $\boldsymbol{y}$ from training set images. This random subsetting of training images into context and target sets transforms this into a meta-learning problem, and the "encoder" $q_\phi(z|\boldsymbol{x}, \boldsymbol{y})$ must learn to generalize over different context set sizes, with less posterior uncertainty as the context set size grows.

**Neural processes**    Our work builds on neural processes (NPs) (Garnelo et al., 2018a;b). NPs are deep neural network conditional generative models. Multiple variants of NPs have been proposed (Garnelo et al., 2018a;b; Kim et al., 2018), and careful empirical comparisons between them appear in the literature (Grover et al., 2019; Le et al., 2018).

NPs employ an alternative training objective to Eq. (2) arising from the fact that the graphical model in Fig. 2 allows a Bayesian update on the distribution of $z$, conditioning on the entire context set to produce a posterior $p_\theta(z|\boldsymbol{x}_\mathcal{C}, \boldsymbol{y}_\mathcal{C})$. If the generative model is in a tractable family that allows analytic updates of this kind, then the NP objective corresponds to maximizing

$$\mathbb{E}_{\boldsymbol{z} \sim q_\phi(z|\boldsymbol{x}_\mathcal{T},\boldsymbol{y}_\mathcal{T})} \left[ \log \frac{p_\theta(\boldsymbol{y}_\mathcal{T}|z,\boldsymbol{x}_\mathcal{T})p_\theta(z|\boldsymbol{x}_\mathcal{C},\boldsymbol{y}_\mathcal{C})}{q_\phi(z|\boldsymbol{x}_\mathcal{T},\boldsymbol{y}_\mathcal{T})} \right] \approx \mathbb{E}_{\boldsymbol{z} \sim q_\phi(z|\boldsymbol{x}_\mathcal{T},\boldsymbol{y}_\mathcal{T})} \left[ \log \frac{p_\theta(\boldsymbol{y}_\mathcal{T}|z,\boldsymbol{x}_\mathcal{T})q_\phi(z|\boldsymbol{x}_\mathcal{C},\boldsymbol{y}_\mathcal{C})}{q_\phi(z|\boldsymbol{x}_\mathcal{T},\boldsymbol{y}_\mathcal{T})} \right] \tag{4}$$

where replacing $p_\theta(z|\boldsymbol{x}_\mathcal{C}, \boldsymbol{y}_\mathcal{C})$ with its variational approximation is typically necessary because most deep neural generative models have a computationally inaccessible posterior. This "NP objective"

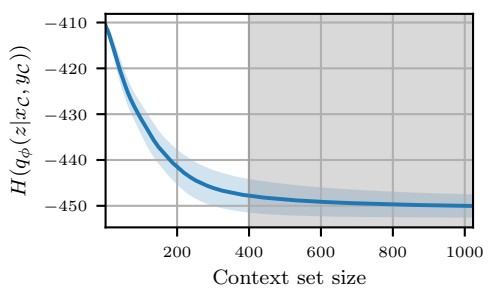 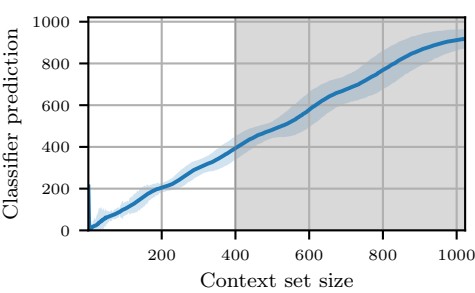

(a) Variational posterior entropy         (b) Classifier prediction

Figure 3: Posterior contraction of $q_\phi(z|\boldsymbol{x}_\mathcal{C}, \boldsymbol{y}_\mathcal{C})$ in a NP+max pooling model. (a) The entropy of $q_\phi(z|\boldsymbol{x}_\mathcal{C}, \boldsymbol{y}_\mathcal{C})$ as a function of context set size, averaged over different tasks (images) and context sets. The gray shaded area in both plots indicates context set sizes that did not appear in the training data for the amortization artifact. (b) Predictions of a classifier trained to infer the context set size given only $s_\mathcal{C}$, the pooled embedding of a context set. Equivalent results for the standard NP+mean pooling encoder and for ANP appear in the Supplementary Material.

can be trained end-to-end, optimizing for both $\phi$ and $\theta$ simultaneously, where the split of training data into context and target sets must vary in terms of context set size. The choice of optimizing Eq. (4) instead of Eq. (2) is largely empirical (Le et al., 2018).

## 3    CALIBRATED UNCERTAINTY

Quantifying and calibrating uncertainty in generative models remains an open problem, particularly in the context of amortized inference. Previous work on uncertainty calibration has focused on problems with relatively simpler structure. For example, in classification and regression problems with a single dataset, prior work framed the problem as predicting a cumulative distribution function that is close to the data-generating distribution, first as a model diagnostic (Gneiting et al., 2007) and subsequently as a post-hoc adjustment to a learned predictor (Kuleshov et al., 2018). A version of the latter approach was also applied to structured prediction problems (Kuleshov & Liang, 2015).

Previous approaches are conceptually similar to our working definition of calibrated uncertainty. However, we seek calibrated uncertainty on a *per-image, per-conditioning set* basis, which is fundamentally different from previous settings. Quantification of all aspects of generative model performance is an area of ongoing research, with uncertainty quantification a particularly challenging problem.

## 4    UNCERTAINTY IN NEURAL PROCESS MODELS

In this section, we investigate how NP models handle uncertainty. A striking property of NP models is that as the size of the (random) context set increases, there is less sampling variation in target samples generated by passing $z \sim q_\phi(z|\boldsymbol{x}_\mathcal{C}, \boldsymbol{y}_\mathcal{C})$ through the decoder. The samples shown in Fig. 1 are the likelihood mean (hence a deterministic function of $z$), and so the reduced sampling variation can only be produced by decreased posterior uncertainty. Our experiments confirm this, as shown in Fig. 3a: posterior uncertainty (as measured by entropy) decreases for increasing context size, *even beyond the maximum training context size*. Such posterior contraction is a well-studied property of classical Bayesian inference and is a consequence of the inductive bias of exchangeable models. However, NP models do not have the same inductive bias explicitly built in. How do trained NP models exhibit posterior contraction without being explicitly designed to do so? How do they learn to do so during training?

A simple hypothesis is that the network somehow transfers the context size through the pooling operation and into $\rho_\phi(s_\mathcal{C})$, which uses that information to set the posterior uncertainty. That hypothesis is supported by Fig. 3b, which shows the results of training a classifier to infer the context size given

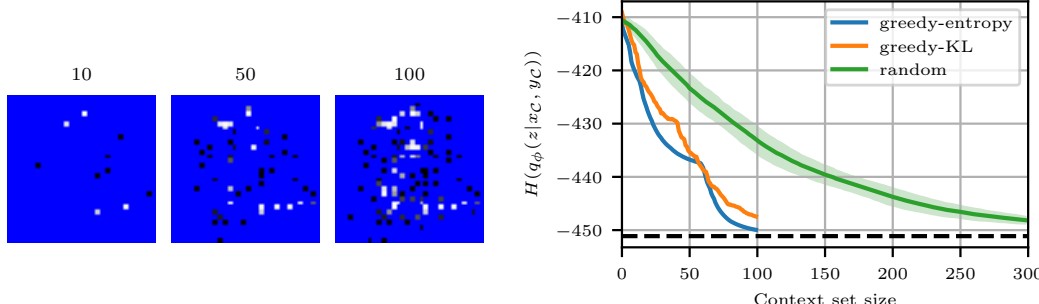

Figure 4: (Left) The first $\{10, 50, 100\}$ pixels greedily chosen to minimize $D_{\text{KL}}(q_\phi(z|\boldsymbol{x}, \boldsymbol{y})||q_\phi(z|\boldsymbol{x}_\mathcal{C}, \boldsymbol{y}_\mathcal{C}))$. These pixels are highly informative about $z$, but only a subset of them will appear in the vast majority of random context sets. (Right) Posterior entropy decreasing as context size increases, for different methods of generating a context set: green is the average over 100 random context sets of each size; blue greedily chooses context pixels to minimize posterior entropy; and orange greedily minimizes $D_{\text{KL}}(q_\phi(z|\boldsymbol{x}, \boldsymbol{y})||q_\phi(z|\boldsymbol{x}_\mathcal{C}, \boldsymbol{y}_\mathcal{C}))$. The black dashed line represents the posterior entropy when conditioned on the full image.

only $s_\mathcal{C}$. However, consider that within a randomly generated context set, some observations are more informative than others. For example, Fig. 4 shows the first $\{10, 50, 100\}$ pixels of an MNIST digit 2, greedily chosen to minimize $D_{\text{KL}}(q_\phi(z|\boldsymbol{x}, \boldsymbol{y})||q_\phi(z|\boldsymbol{x}_\mathcal{C}, \boldsymbol{y}_\mathcal{C}))$. If $z$ is interpreted to represent, amongst other things, which digit the image contains, then a small subset of pixels determine which digits are possible.

It is these highly informative pixels that drive posterior contraction in a trained NP. In a random context set, the number of highly informative pixels is random. For example, a max-pooled embedding saturates with the $M$ most highly informative context pixels, where $M \leq d$, the dimension of embedding space. On average, a random context set of size $n$, taken from an image with $N$ pixels, will contain only $nM/N$ of the informative pixels. In truth, Fig. 3 displays how the information content of a context depends, on average, on the size of that context. Indeed, greedily choosing context pixels results in much faster contraction (Fig. 4).

**Learning to contract** Posterior contraction is implicitly encouraged by the NP objective Eq. (4). It can be rewritten as

$$\mathbb{E}_{\boldsymbol{z} \sim q_\phi(z|\boldsymbol{x}_\mathcal{T}, \boldsymbol{y}_\mathcal{T})} \left[ \log p_\theta(\boldsymbol{y}_\mathcal{T}|z, \boldsymbol{x}_\mathcal{T}) \right] - D_{\text{KL}}(q_\phi(z|\boldsymbol{x}_\mathcal{T}, \boldsymbol{y}_\mathcal{T})||q_\phi(z|\boldsymbol{x}_\mathcal{C}, \boldsymbol{y}_\mathcal{C})) . \tag{5}$$

The first term encourages perfect reconstruction of $y_\mathcal{T}$, and discourages large variations in $z \sim q_\phi(z|\boldsymbol{x}_\mathcal{T}, \boldsymbol{y}_\mathcal{T})$, which would result in large variations in predictive log-likelihood. This effect is stronger for larger target sets since there are more target pixels to predict. In practice, $\mathcal{C} \subset \mathcal{T}$, so the first term also (indirectly) encourages posterior contraction for increasing context sizes. The second term, $D_{\text{KL}}(q_\phi(z|\boldsymbol{x}_\mathcal{T}, \boldsymbol{y}_\mathcal{T})||q_\phi(z|\boldsymbol{x}_\mathcal{C}, \boldsymbol{y}_\mathcal{C}))$, reinforces the contraction by encouraging the context posterior to be close to the target posterior.

Although the objective encourages posterior contraction, the network mechanisms for achieving contraction are not immediately clear. Ultimately, the details depend on interplay between the pixel embedding function, $h_\phi$, the pooling operation $\oplus$, and $\rho_\phi$. We focus on mean and max pooling.

**Max pooling** As the size of the context set increases, the max-pooled embedding $s_\mathcal{C} = \oplus_{i=1}^n s_i$ is non-decreasing in $n$; in a trained NP model, $||s_\mathcal{C}||$ will increase each time an informative pixel is added to the context set; it will continue increasing until the context embedding saturates at the full image embedding. At a high level, this property of max-pooling means that the $\sigma_\mathcal{C}$ component of $\rho_\phi(s_\mathcal{C})$ has a relatively simple task: represent a function such that the posterior entropy is a decreasing function of all dimensions of the embedding space. An empirical demonstration that $\rho_\phi$ achieves this can be found in the Supplementary Material.

**Mean pooling** For a fixed image, as the size of a random context set increases, its mean-pooled embedding will, on average, become closer to the full image embedding. Moreover, the mean-pooled embeddings of all possible context sets of the image are contained in the convex set whose

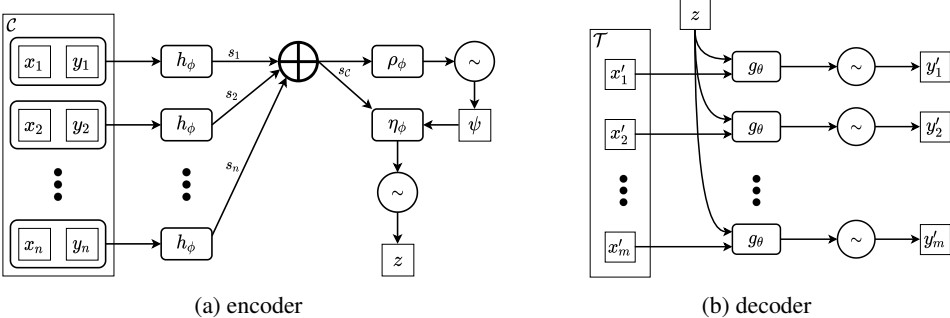

(a) encoder                                          (b) decoder

Figure 5: Our modified neural process architecture. The encoder produces a permutation invariant embedding that parameterizes a stochastic task encoding $z$ as follows: features extracted from each element of the context set using neural net $h_\phi$ are pooled, then passed to other neural networks $\rho_\phi$ and $\eta_\phi$ that control the distribution over task embedding $z$. The decoder uses such a task encoding along with embeddings of target inputs to produce the output distribution for each target input.

hull is formed by (a subset of) the individual pixel embeddings. The $\sigma_{\mathcal{C}}$ component of $\rho_\phi(s_{\mathcal{C}})$, then, must approximate a function such that the posterior entropy is a convex function on the convex set formed by individual pixel embeddings, with minimum at or near the full image embedding. Learning such a function across the embeddings of many training images seems a much harder learning task than that required by max pooling, which may explain the better performance of max pooling relative to mean pooling in NPs (see Section 6).

**Generalizing posterior contraction**   Remarkably, trained NP-based models generalize their posterior contraction to context and target sizes not seen during training (see Fig. 3). The discussion of posterior contraction in NPs using mean and max pooling in the previous paragraphs highlights a shared property: for both models, the pooled embeddings of all possible context sets that can be obtained from an image are in a convex set that is determined by a subset of possible context set embeddings. For max-pooling, the convex set is formed by the max-pooled embedding of the $M$ "activation" pixels. For mean-pooling, the convex set is obtained from the convex hull of the individual pixel embeddings. Furthermore, the full image embedding in both cases is contained in the convex set. We conjecture that a sufficient condition for an NP image completion model to yield posterior contraction that generalizes to context sets of unseen size is as follows: For any image, the pooled embedding of every possible context set (which includes the full image) lies in a convex subset of the embedding space.

## 5   NETWORK ARCHITECTURE

The network architectures we employ for our experiments build on NPs, inspired by our findings from Section 4. We describe them in detail in this section.

**Encoder**   The encoder $q_\phi(z|\boldsymbol{x}_{\mathcal{C}}, \boldsymbol{y}_{\mathcal{C}})$ takes input observations from an i.i.d. model (see Fig. 2, plate over $\mathcal{C}$), and therefore its encoding of those observations must be permutation invariant if it is to be learned efficiently. Our $q_\phi$, as in related NP work, has a permutation-invariant architecture,

$$s_i = h_\phi(x_i, y_i),\ 1 \leq i \leq n; \quad s_{\mathcal{C}} = \oplus_{i=1}^n s_i; \quad (\mu_{\mathcal{C}}, \sigma_{\mathcal{C}}) = \rho_\phi(s_{\mathcal{C}}); \quad q_\phi(z|\boldsymbol{x}_{\mathcal{C}}, \boldsymbol{y}_{\mathcal{C}}) = \mathcal{N}(\mu_{\mathcal{C}}, \sigma_{\mathcal{C}}^2)\ .$$

Here $\rho_\phi$ and $h_\phi$ are neural networks and $\oplus$ is a permutation-invariant pooling operator. Fig. 5 contains diagrams of a generalization of this encoder architecture (see below). The standard NP architecture uses mean pooling; motivated by our findings in Section 4, we also employ max pooling.

**Hierarchical Variational Inference**   In order to achieve better calibrated uncertainty in small context size regimes, a more flexible approximate posterior could be beneficial. tConsider the MNIST experiment shown in Fig. 6. Intuitively, an encoder could learn to map from the context set to a one-dimensional discrete $z$ value that lends support only to those digits that are compatible with the context pixel values at the given context pixel locations ($\boldsymbol{x}_{\mathcal{C}}, \boldsymbol{y}_{\mathcal{C}}$). This suggests that $q_\phi$ should be flexible enough to produce a multimodal distribution over $z$, which can be encouraged by making $q_\phi$ a mixture and corresponds to a hierarchical variational distribution (Ranganath et al., 2016;

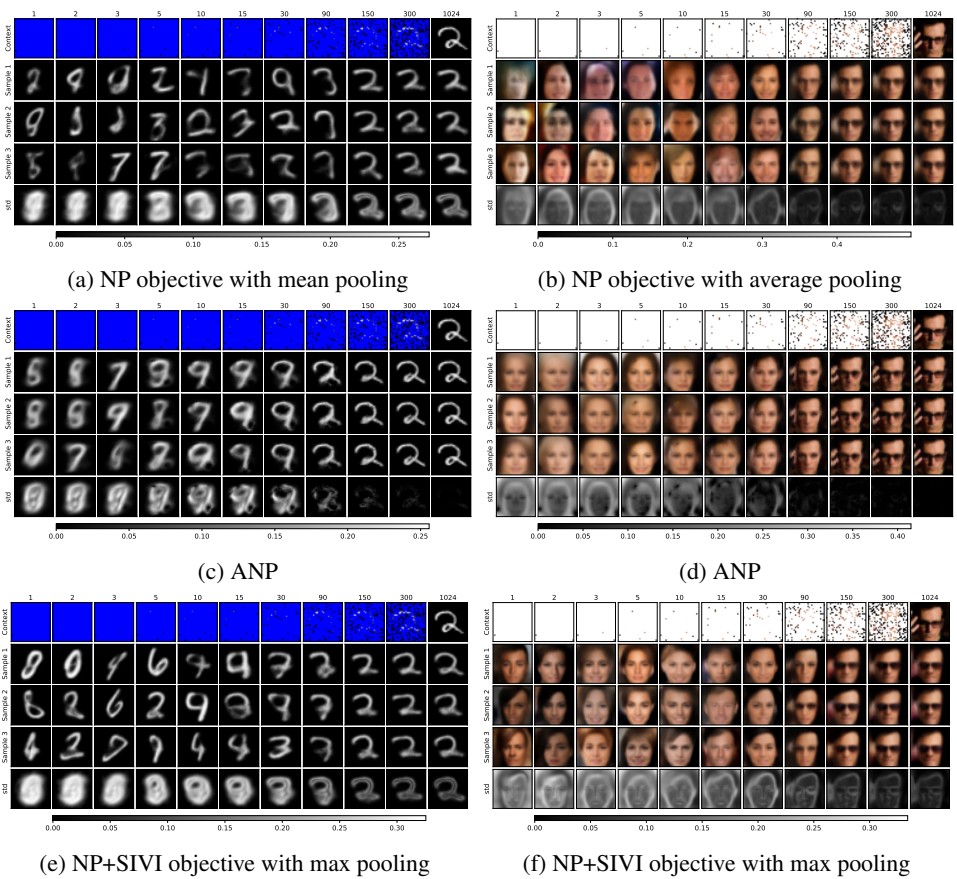

(a) NP objective with mean pooling        (b) NP objective with average pooling

(c) ANP                          (d) ANP

(e) NP+SIVI objective with max pooling     (f) NP+SIVI objective with max pooling

Figure 6: Example MNIST and CelebA image completion tasks, for each of three NP methods. The following guide applies to each block. The top row shows context sets of different sizes (context sets are exactly the same for all methods), i.e., one task per column. The ground truth image is in the upper right corner. The rows correspond to the mean function produced by $g_\theta$ for different sampled values of $z$. The bottom row shows an empirical estimate of the standard deviation of the mean function from 1000 draws of $z$, a direct visualization of the uncertainty encoding.

Yin & Zhou, 2018; Sobolev & Vetrov, 2019). Specifically, the encoder structure described above, augmented with a mixture variable is

$$q_\phi(z|\boldsymbol{x}, \boldsymbol{y}) = \int q_\phi(\psi|\boldsymbol{x}, \boldsymbol{y}) q_\phi(z|\psi, \boldsymbol{x}, \boldsymbol{y}) d\psi \ . \tag{6}$$

This is shown in Fig. 5. For parameter-learning, the ELBO in Eq. (2) is targeted. However, the hierarchical structure of the encoder makes this objective intractable. Therefore, a tractable lower bound to the ELBO is used as the objective instead. In particular, the objective is based on semi-implicit variational inference (SIVI) (See Appendix A.3).

**Decoder** The deep neural network stochastic decoder in our work is standard and not a focus. Like other NP work, the data generating conditional likelihood in our decoder is assumed to factorize in a conditionally independent way, $p_\theta(\boldsymbol{y}_{\mathcal{T}}|z, \boldsymbol{x}_{\mathcal{T}}) = \prod_{i=1}^{m} p_\theta(y_i'|z, x_i')$, where $m$ is the size of the target set and $x_i'$ and $y_i'$ are a target set input and output respectively. Fig. 5b shows the decoder architecture, with the neural network $g_\theta$ the link function to a per pixel likelihood.

## 6 EXPERIMENTAL EVALUATION

We follow the experimental setup of Garnelo et al. (2018b), where images are interpreted as functions that map pixel locations to color values, and image in-painting is framed as an amortized

| Method | MNIST | FashionMNIST | CelebA |
|---|---|---|---|
| NP+mean | $0.96 \pm 0.12$ | $0.93 \pm 0.15$ | $2.91 \pm 0.30$ |
| ANP+mean | $0.55 \pm 0.12$ | $0.57 \pm 0.11$ | $1.81 \pm 0.18$ |
| NP+max | $1.07 \pm 0.11$ | $1.02 \pm 0.19$ | $3.17 \pm 0.30$ |
| SIVI+max | $0.99 \pm 0.25$ | $0.96 \pm 0.16$ | $2.99 \pm 0.39$ |

Table 1: Predictive held-out test log-likelihood

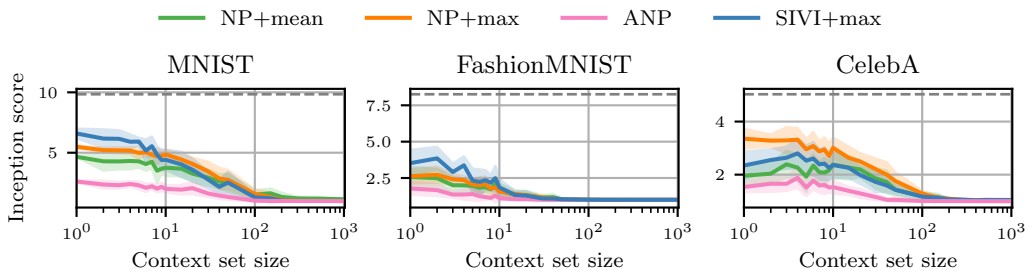

Figure 7: Inception scores of conditional samples.

predictive inference task where the latent image-specific regression function needs to be inferred from a small context set of provided pixel values and locations. For ease of comparison to prior work, we use the same MNIST (LeCun et al., 1998) and CelebA (Liu et al., 2015) datasets. Additionally, we run an experiment on FashionMNIST dataset (Xiao et al., 2017). Specific architecture details for all networks are provided in Appendix A and open-source code for all experiments will be released at the time of publication.

**Qualitative Results** Fig. 6 shows qualitative image in-painting results for MNIST and CelebA images. Qualitative results for FashionMNIST are shown in Appendix D. It is apparent in all three contexts that ANPs perform poorly when the context set is small, despite the superior sharpness of their reconstructions when given large context sets. The sets of digits and faces that ANPs produce are not sharp, realistic, nor diverse. On the other hand, their predecessor, NP (with mean pooling), arguably exhibits more diversity but suffers at all context sizes in terms of realism of the images. Our NP+SIVI with max pooling approach produces results with two important characteristics: 1) the images generated with a small amount of contextual information are sharper and more realistic; and 2) there is high context-set-compatible variability across the i.i.d. samples. These qualitative results demonstrate that max pooling plus the SIVI objective result in posterior mean functions that are sharper and more appropriately diverse, except in the high context set size regime where diversity does not matter and ANP produces much sharper images. Space limitations prohibit showing large collections of samples where the qualitative differences are even more readily apparent. Appendix L contains more comprehensive examples with greater numbers of samples.

**Quantitative Results** Quantitatively assessing posterior predictive calibration is an open problem (Salimans et al., 2016; Heusel et al., 2017). Table 1 reports, for the different architectures we consider, predictive held out test-data log-likelihoods averaged over 10,000 MNIST, 10,000 FashionMNIST and 19,962 CelebA test images respectively. While the reported results make it clear that max pooling improves held-out test likelihood, likelihood alone does not provide a direct measure of sample quality nor diversity. It simply measures how much mass is put on each ground-truth completion. It is also important to note that in our implementation of ANP, in contrast to its original paper, the observation variance is fixed and that is why ANP performs poorly in Table 1. An ANP model with learned observation variance outperforms all the other models in terms of held-out test likelihood. However, it is empirically shown that learning the observation variance in NP models with a deterministic path (including ANPs) hurts the diversity of generated samples (Le et al., 2018) (see Appendix C for a detailed discussion and additional results for ANP model with learned variance).

Borrowing from the generative adversarial networks community, who have faced the similar problems of how to quantitatively evaluate models via examination of the samples they generate, we

compute inception scores (Salimans et al., 2016) using conditionally generated samples for different context set sizes for all of the considered NP architectures and report them in Fig. 7. Inception score is the mutual information between the generated images and their class labels predicted by a classifier, in particular, inception network (Szegedy et al., 2016). However, since inception network is an ImageNet (Deng et al., 2009) classifier, it is known to lead to misleading inception scores when applied to other image domains (Barratt & Sharma, 2018). We therefore use trained MNIST, FashionMNIST, and CelebA classifiers in place of inception network (He et al., 2016). (See Appendix H for details.) The images used to create the results in Fig. 7 are the same as in Figs. 6 and 11. For each context set size, the reported inception scores are aggregated over 10 different randomly chosen context sets. The dark gray dashed lines are the inception scores of training samples and represent the maximum one might hope to achieve at a context set size of zero (these plots start at one).

For small context sets, an optimally calibrated model should have high uncertainty and therefore generate samples with high diversity, resulting in high inception scores as observed. As the context set grows, sample diversity should be reduced, resulting in lower scores. Here again, architectures using max pooling produce large gains in inception score in low-context size settings. Whether the addition of SIVI is helpful is less clear here (see Appendix I for a discussion on the addition of SIVI). Nonetheless, the inception score is again only correlated with the qualitative gains we observe in Fig. 6.

## 7 CONCLUSION

The contributions we report in this paper include suggested neural process architectures (max pooling, no deterministic path) and objectives (regular amortized inference versus the heuristic NP objective, SIVI versus non-mixture variational family) that produce qualitatively better calibrated posteriors, particularly in low context cardinality settings. We provide empirical evidence of how natural posterior contraction may be facilitated by the neural process architecture. Finally, we establish quantitative evidence that shows improvements in neural process posterior predictive performance and highlight the need for better metrics for quantitatively evaluating posterior calibration.

We remind the reader that this work, like most other deep learning work, highlights the impact of varying only a small subset of the dimensions of architecture and objective degrees of freedom. We found that, for instance, simply making $\rho_\phi$ deeper than that reported in the literature improved baseline results substantially. The choice of learning rate also had a large impact on the relative gap between the reported alternatives. We report what we believe to be the most robust configuration across all the configurations that we explored: max pooling and SIVI consistently improve performance.

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
