# OpenReview forum: "Uncertainty in Neural Processes"
_ICLR.cc/2021/Conference — Reject_

### Official Review · AnonReviewer1 · 2020-10-26
**Interesting idea, but the evaluation is not quite thorough enough**

**Rating:** 5
**Confidence:** 4

**Review:**

The authors propose an extension to Neural Processes (NPs), where they use max-pooling instead of mean-pooling as the aggregation function and use a mixture distribution with hierarchical VI for the decoder. They show that this slightly improves the predictive likelihoods, but crucially strongly improves the diversity of posterior samples when the conditioning set is small, which they argue is an important feature of such models.

Major comments:
- What is the actual impact of using the SIVI? In most experiments, it looks like NP+max performs just as well.
- What would happen if one would use mean-pooling AND max-pooling and just concatenate the two to yield an aggregated representation? Wouldn't that combine the best of both worlds and the downstream decoder could learn which representation to use for the mean and the variance prediction?
- Could these ideas (SIVI, max-pooling) also be combined with more modern NP architectures (like Attentive NPs or Convolutional NPs)?

Minor comments:
- It is argued that the max-pooling is naturally better at capturing useful information for estimating the posterior variances. But what about the posterior mean? Shouldn't the mean-pooling be better for that?
- In Tab. 1, "NP+max" seems to be the best-performing model. Why is it not shown in Tab. 6?

Summary:
I think the focus on the diversity of posterior samples is very interesting and highlights some important property of these kinds of models. However, given the relative simplicity of the proposed extensions, I feel like they are not studied extensively enough. For the paper to provide a clear value for the community, I think it would be good to extend the experiments to cover the whole combination space of {NP, ANP, SIVI} x {mean-pooling, max-pooling, mean+max-pooling}, so that it becomes clearer what the influence of the different design choices are in combination with each other.

---

> ### Author Response · Authors · 2020-11-23
> **Response to reviewer 1**
>
> We thank the reviewer for the thorough comments. Here is our response:
>
> __SIVI__: As mentioned in our response to _AnonReviewer2_, we have added additional results to the supplementary material investigating the effect of SIVI. Furthermore, we have added a new experiment on FashionMNIST supporting our hypothesis that a 1-level hierarchical encoder achieves more performance gain on a dataset with a finite set of well-separated classes. Also, our inception scores are more reliable on such datasets.
>
> __Mixed pooling__: The idea of using both max and mean pooling is interesting and we have actually tried it previously, but it did not perform better than the models with only max pooling. We have added inception scores plots to the appendices (see Appendix J)
>
> __ANP and ConvNP__: We have done experiments with ANP with max pooling and ANP with SIVI and max pooling, but these do not mitigate the lack of sample diversity in ANP. We are not sure of how Convolutional NPs would perform. We have added this statement to the paper.
>
> __Max pooling and posterior mean__: We argue that in the case of small conditioning sets, the posterior uncertainty is more relevant than the posterior mean. With large conditioning sets, the posterior mean begins to dominate. Even in the case of large conditioning sets, our experimental results suggest that max pooling does not do worse (if not better) than mean pooling in capturing the posterior mean (most evident comparing the samples in the right-most columns of Fig. 6a and Fig. 6e where the SIVI with max pooling samples resemble the original image more closely than the mean pooling samples).
>
> __NP+max results__: For space constraints, Fig. 6 only shows the "standard" models i.e. models proposed in the previous works. Qualitative results from NP+max are reported in Appendix L, and results from some other non-standard configurations are reported in the Appendix C.

---

### Official Review · AnonReviewer3 · 2020-10-27
**This paper proposes an improvement of the standard NP, and the authors have investigated the posterior contraction of NP.**

**Rating:** 8
**Confidence:** 4

**Review:**

This paper proposes an improvement of the standard NP by using a mixture distribution \q_{\phi}, semi-implicit variational inference, and max pooling to capture the multimodel structure of the posterior distribution. Replacing one normal Gaussian distribution with a mixture (of Gaussians, normally) is a widely-adopted idea in latent variable models including NP; the adopted semi-implicit variational inference was originally developed in Yin and Zhou ICML 2018, and no further improvement on this inference method is proposed in this manuscript; max pooling is one of three commonly used pooling methods, i.e., max, min, and mean pooling. using one of them to replace another is simple but the explanation of the reason why max pooling is better is interesting and profound. So, the improvement is weak although it is shown to be effective by the empirical study. More importantly, the authors have investigated the posterior contraction of NP. It is interesting. The relationship between the two parts of the objective function of NP has been discussed related to the posterior contraction, both parts have contributed to the contraction apart from their classical explanation on reconstruction and regularization. To my best knowledge, it is the first work to discuss the posterior contraction of NP. It is a classical property in Bayesian and this link will enable further theoretical analysis for NP.

---

> ### Author Response · Authors · 2020-11-23
> **Response to reviewer 3**
>
> We thank the reviewer for the positive review and highlighting what they found to be conceptually interesting.

---

### Official Review · AnonReviewer2 · 2020-10-27
**This paper proposes to improve neural processes by changing the aggregation operator and using a hierarchical latent structure. These decisions are aimed at improving the performance in the low-data regime.**

**Rating:** 5
**Confidence:** 3

**Review:**

The paper aims at increasing the sample diversity of neural processes when the condition set is small, while maintaining visual fidelity. The low-data regime is arguably where neural processes are most interesting, and in that regard the paper is right to turn to this setting. The discussion on how different aggregation functions affect the predictive uncertainty of the neural process is also appreciated, as is the experiment on regressing the size of the condition set based on the latent embedding.

Unfortunately, the experiments section does not paint a clear enough picture. While the experiments show us that the proposed modifications have some benefits, it is not so clear how much each part contributes. Especially the contribution of the SIVI bound is hard to judge. As it stands the paper feels a bit incomplete in this regard. For that reason I cannot recommend accepting the paper at this stage, though I am willing to revise my score based on the authors' response. Specifically, I'd appreciate if the paper could make a clear case for adopting the hierarchical latent variable structure and the SIVI bound, as these add complexity to the method (while the max-aggregator does not).

### Pros

* The paper deals with a relevant issue. Neural processes are most interesting when the condition set is small, and this scenario has so far been largely ignored.
* The discussion on the choice of aggregator is useful, as are the experiments on the variational posterior entropy and the prediction of the context set size.

### Cons

* It is unclear how much each modification (SIVI and max-pooling) contribute. The experimental results compare SIVI+max pooling with NP+max pooling, but SIVI+mean is omitted. It's also notable that NP+max seems to work better than SIVI+max in the CelebA dataset. Some discussion would be helpful here, as I don't see any reason why a hierarchical latent structure should hurt in any case, barring optimization difficulties.
* I am not familiar with SIVI and I don't expect the average reader to be either. I'd appreciate some discussion on the choice of using it.

### Other comments

* The inception score sounds like the mutual information between the class label and the generated image. I expect that stating this would help some readers.
* Perhaps it would help to look at each component in isolation and in different settings, e.g. outside the image domain. I can see how max-pooling might be good for images, while other aggregation methods might have an edge in, e.g. a dataset of robot joint trajectories. Other people have investigated the choice of aggregation method, and reading this work reminded me of work by [Soelch et al.](https://arxiv.org/abs/1903.07348), which might be interesting to the authors.

---

> ### Author Response · Authors · 2020-11-23
> **Response to reviewer 2**
>
> We appreciate the reviewer’s thoughtful comments and questions. Here is our response:
>
> __Contribution of SIVI__: To see the effect of hierarchical proposals in isolation, it is not enough to compare SIVI+max with NP+max or SIVI+mean with NP+mean, because SIVI models optimize a lower bound to $p(y_\mathcal{T} | x_\mathcal{T})$ (see Eq. 2 in the paper) while NP models target an approximation to the lower bound to $p(y_\mathcal{T} | x_\mathcal{T}, x_\mathcal{C}, y_\mathcal{C})$. Therefore, we should compare SIVI models with models that optimize Eq. 2, where SIVI outperforms the baseline on both datasets. We argue that although the NP objective empirically achieves better results, the additional approximation in its objective (that replaces $p(z|x_\mathcal{C}, y_\mathcal{C})$ with $q(z|x_\mathcal{C}, y_\mathcal{C})$) has not yet been justified theoretically. We have added a section to the appendices where the effect of SIVI is discussed in more detail and inception score plots for different models are provided.
>
> Additionally, we have added an additional experiment on FashionMNIST and added the results to the paper. SIVI+max outperforms the other methods on this dataset as well. We conjecture that SIVI achieves a bigger performance gain when the dataset has a finite set of well-separated classes, like MNIST and FashionMNIST. This means that perhaps SIVI is able to better separate the different classes in the latent space. In addition, since we train classifiers on each dataset to compute inception score, having such well-behaved datasets results in more reliable trained classifiers and more accurate inception scores. It is possible that on datasets without well-defined classes like CelebA, a hierarchical encoder with more levels of hierarchy might be able to capture the multi-modality of the posterior, though with an additional approximation of the ELBO with each level of additional hierarchy.
>
> We completely agree with the reviewer regarding the other suggestions and have updated the paper to include the mutual information view of inception score and choice of SIVI.
>
> We thank the reviewer again for the helpful comments and related work suggestions.

---

> > ### Comment · AnonReviewer2 · 2020-11-24
> > **Reply to the authors**
> >
> > Thank you very much for your reply. I will confess that I did not realize the SIVI models were optimizing a different bound, this is on me, as it is stated in the text (though perhaps it should be highlighted more).
> >
> > I also appreciate the additional experiments on FashionMNIST, which strengthen the case for the proposed method. Still, I continue to have the feeling that there are some crucial points in this paper which should be clarified further:
> >
> > * Why do the results in CelebA contradict those in the other two datasets? Is this because the underlying classifier is under-performing because of a lack of clear class structure, or is it because SIVI is somehow less well-suited to problems where there is no clear class structure?
> > * If SIVI is less well-suited to problems like CelebA, why is that the case? Why should a hierarchical latent variable structure hurt in any case, barring optimization issues?
> >
> > I also have the feeling that too many modifications are being presented at the same time and not all are receiving the same amount of attention. There is a) the max-aggregator, which is motivated by an interesting discussion in section 4, b) the hierarchical latent variable structure, which is motivated by the idea that a more flexible variational distribution will improve things and c) the idea to optimize eq. 2 instead of eq. 4. Of these, I think the paper is doing a good job at motivating a), while b) and c) are less clear to me. To conclude, what would change this paper from "marginally below acceptance threshold" to a "clear accept" in my opinion is:
> >
> > * Resolve the issues on CelebA.
> > * Flesh out the ideas on b) and c), giving a clear motivation for why these are necessary or beneficial, or provide a clear experimental signal (which currently is clouded by the CelebA results).
> > * Discuss alternative forms of flexible posteriors for b). A hierarchical structure is only one way of achieving this. If there is a need for increased flexibility, other approaches such as normalizing flows would come into question. For this point, the authors can look at work by [Cremer et al.](https://arxiv.org/pdf/1801.03558.pdf]) which discusses the effect of flexibility on posterior inference in the context of VAEs.
> > * Discuss the implications of changing the objective. For instance, section 4 states that the NP objective encourages posterior contraction. How does this behave when using eq. 2?

---

> > > ### Author Response · Authors · 2020-11-25
> > > **Reply to reviewer 2**
> > >
> > > Thank you for the follow-up and raising insightful questions and concerns.
> > >
> > > Regarding (b): The hierarchical encoder should be compared to models that optimize Eq. 2 in the paper (which we refer to as the ELBO objective), and the results in Appendix I show that SIVI models indeed outperform their ELBO counterparts. These results provide empirical evidence that the hierarchical encoder should be beneficial. It is also because of these results that we expect the overall performance drop for CelebA is due to other factors (in particular, the ELBO objective). The results in Appendix I do appear to support this hypothesis as the hierarchical encoder model at least achieves the best results among the ELBO models, but overall the models with the NP objective perform better than the models with the ELBO objective.
> > >
> > > Regarding (c): We argue that because of the lack of theoretical justification for the additional approximation in the NP objective, the better-known ELBO objective may be more desirable despite its worse empirical performance in some experiments. It is possible that a model which uses the NP objective with a hierarchical encoder would empirically perform better than NP+max, though a tractable objective would first need to be derived and this is better left as future work.
> > >
> > > We agree with the reviewer that it is important to figure out what exactly causes our proposed model to perform worse than NP+max on CelebA. Possible sources that we are currently aware of include
> > > (regarding the inception score results) the underlying classifier used to compute the inception score, as the classifier does not perform as well as the MNIST and FashionMNIST classifiers. Related questions include whether the classifier was well-trained in the first place, what happens if the number of classes is changed, etc. There is also the bigger question of whether the inception score/GAN objective is actually an appropriate measure of what we desire in a model.
> > > complexity of the CelebA latent space. More levels to the hierarchy or a more expressive variational family may be required. There are many other options for obtaining a more expressive variational family (including the mentioned normalizing flows and their conditional variants).
> > > Given that this list is already non-trivial and that there are likely other sources to consider, we believe a deeper investigation of the cause is better suited for follow-up work.
> > >
> > > Regarding the implications of using Eq. 2: as mentioned in Section 4, the NP objective can be written as two terms that both encourage posterior contraction. It is true that Eq. 2 loses the interpretation of the second term (the KL between the posterior conditioned on the target and the posterior conditioned on the context). However, we stress that these are merely interpretations of the objective that help explain why contraction occurs. It is clear from all of our experiments that posterior contraction still occurs with Eq. 2 as the objective. It is just that the objective does not provide an intuitive explanation for the contraction in this case.

---

### Official Review · AnonReviewer4 · 2020-10-29
**contributions not clear**

**Rating:** 5
**Confidence:** 4

**Review:**

Summary: This paper is an empirical investigation into the role of architecture and objective choices in Neural Process (NP) models when the amount of conditioning data is limited. Specifically, they investigate the question of well-calibrated uncertainty.

Clarity: The overall quality of the writing is clear, but missing details/explanations leave some claims unjustified and therefore lines of argument difficult to follow.

Originality: Limited -- the paper is mostly an empirical investigation, with the modifications to existing NPs being (a) max-pooling and (b) SIVI on the posterior z’s.

Significance: Seems limited (see more on “Cons” section and “Questions/comments”), though I am wondering whether the authors could have done more to emphasize the main contributions of this work. The paper’s significance for me has been hampered by lack of details and clear takeaway/implications of the results.

Pros: There’s been a lot of work on NPs and their variants, but less so on empirically investigating how and when they work well. There is also a range of systematic empirical evaluations both in the main text and supplement, which I appreciated.

Cons: There are some assumptions/statements that would be beneficial to elaborate upon in the paper. First, calibrated uncertainty is defined to be “high sample diversity for small conditioning sets; and sharp-looking, realistic images for any size of conditioning set.” This statement, introduced early on in the paper without much justification, is a point that the authors repeatedly return to, and I found myself wondering why (especially since NPs are built for prediction/regression, so a lot of the prior work on calibration for classification models should hold here such as (Murphy 1973), (Gneiting et. al 2007), (Kuleshov et al., 2018)). What is the benefit of reasoning about calibration in the generative modeling case (e.g. with Inception Scores)?

Additionally, why should a more flexible approximate posterior be more beneficial for better calibrated uncertainty? (Is this a log-likelihood argument, since the log-likelihood decomposes to calibration + “sharpness”?) More broadly, my biggest problem was that the paper makes claims about improving calibration (e.g. samples are obtained from “well calibrated posteriors” in Figure 6) without formally defining how to evaluate “good calibration,” what it means to have “calibrated uncertainty,” etc. I would appreciate if the authors could clear up any misunderstandings I may have had about the work.

Questions/comments:
- (Heusel et. al 2017) show that the inception score (IS) is not a valid metric for CelebA -- would the authors report FID scores instead?
- Regarding the comment in Section 4 about posterior contraction: aren’t NPs exchangeable by design via the iid latent variable z (Korshunova et. al 2019, among many others)? I thought that invoking (conditional) de Finetti via the iid latent variable is what allows NPs to model (exchangeable) stochastic processes.
- It would be helpful to formally define “posterior contraction” for the reader -- is it referring to a reduction in posterior uncertainty?
- Intuitively, why would sample diversity decrease as the size of the conditioning set grows? For example, if I have a dataset of 10 examples of only cats and dogs and increase its size to 1000 (which say, also includes examples of sheep), shouldn’t that also increase my sample diversity as well?
- Shouldn’t the arrow going from z -> y_i in Figure 2 be reversed?

-------------
UPDATE: I have read the authors' rebuttal and revised draft and have raised my score to a 5.

---

> ### Author Response · Authors · 2020-11-23
> **Response to reviewer 4**
>
> We would like to thank the reviewer for the comments and constructive criticism. Here are our responses to your concerns:
>
> __Calibrated uncertainty__: The definition of “calibrated uncertainty” in the papers that you listed is conceptually the same as what we use, i.e., that predictive uncertainty should be consistent with the data generating distribution (and using the empirical distribution as a proxy). However, the problem considered here is fundamentally different (and more challenging): we seek calibrated uncertainty on a per-image, per-conditioning set basis, which renders the approaches in those papers either inapplicable or infeasible. GAN-type evaluations of model performance are not perfect, but they are more appropriate for this problem. Progress in this area is important, but not the main point of the paper.
>
> Thanks for pointing out this source of confusion; we have added a short background section discussing previous work on uncertainty calibration.
>
>
> __Contributions__: To clarify the confusion about the contributions of our work, we state our two main contributions here. First, we identify a key performance limitation of neural processes and explore the nature of that limitation empirically and conceptually. Specifically, we investigate how uncertainty is represented in neural process models as a function of the size of the context set. Second, we explore architecture and objective choices, and empirically show which choices lead to models that have better calibrated uncertainty.
>
> __FID__: Regarding your question about IS vs. FID, FID measures how similar two sets of (real and generated) images are. Therefore, FID requires access to “real” images. It is easily applicable to GANs where the goal is to learn the distribution of real objects from which a large dataset of i.i.d. samples is available. However, in our experiments, the “real” data distribution is a conditional distribution and we do not have access to samples from it. However, IS is computed purely from the generated samples and hence is more practical in our setting.
>
> Additionally, as mentioned in the paper, we understand that the standard inception score gives misleading results on CelebA or MNIST, but that is because the standard score is computed using a network trained on ImageNet. That is why we compute the scores using classification networks trained on CelebA/MNIST images.
>
> __Posterior contraction__: As mentioned in the paper, “posterior uncertainty (as measured by entropy) decreases for increasing context size” is what we mean by posterior contraction. This is standard terminology in Bayesian statistics.
>
> __Exchangeability of NPs__: Unfortunately, there is confusion in the NP literature about stochastic processes and the so-called “conditional de Finetti’s theorem.” Much of the confusion seems to stem from an incorrect interpretation that the consistency requirements of Kolmogorov’s Extension Theorem are the same as exchangeability. They are not the same. (As a simple check, note that an infinite-length discrete-time Markov chain is a properly defined stochastic process whose existence is guaranteed by Kolmogorov’s Extension Theorem, but it is not exchangeable.) The upshot is that NP models are *constructed* to have some useful conditional independence properties that ultimately are the same as those *assumed* in Gaussian Process regression (i.e., conditionally independent observations given a random function), but they are not exchangeable.
>
>
> __Graphical model__: Fig. 2 is the generative graphical model for both context and target sets. That is why the arrow is from z -> y_i. We have updated the figure caption to make it clear.

---

### Decision · Program_Chairs · 2021-01-07
**Final Decision**

**Decision:**

Reject

**Comment:**

This paper analyzes some design choices for neural processes, paying particular attention to their small-data performance, uncertainty, and posterior contraction.  This is certainly a worthwhile project, and R3 found the analysis interesting, giving the paper a score of 8.  However, R1, R2, and R4 found the experimental validation to be incomplete and insufficient to support the paper's broader recommendations.  As the paper is investigating the various combinations of implementations, I tend to agree with R1, R2, and R4 that this paper---while having some interesting ideas---needs a bit more precision and breadth to its experiments.